# Screening of targeted panel genes in Brazilian patients with primary ovarian insufficiency

**Monica M. França**[1¤]*, **Mariana F. A. Funari**[1], **Antonio M. Lerario**[2], **Mariza G. Santos**[1], **Mirian Y. Nishi**[1,3], **Sorahia Domenice**[1], **Daniela R. Moraes**[1], **Everlayny F. Costalonga**[4], **Gustavo A. R. Maciel**[5], **Andrea T. Maciel-Guerra**[6], **Gil Guerra-Junior**[7], **Berenice B. Mendonca**[1,3]

1 Unidade de Endocrinologia do Desenvolvimento, Laboratório de Hormônios e Genética Molecular/LIM42, Hospital das Clínicas HCFMUSP, Faculdade de Medicina, Universidade de São Paulo, São Paulo, SP, Brazil, 2 Department of Internal Medicine, Division of Metabolism, Endocrinology and Diabetes, University of Michigan, Ann Arbor, MI, United States of America, 3 Laboratório de Sequenciamento em Larga Escala (SELA), Faculdade de Medicina da Universidade de São Paulo FMUSP, São Paulo, SP, Brazil, 4 Departamento de Clínica Médica, Faculdade de Medicina da Universidade Federal do Espírito Santo, Vitória, Espírito Santo, Brazil, 5 Disciplina de Ginecologia, Departamento de Obstetrícia e Ginecologia Hospital das Clinicas HCFMUSP, Faculdade de Medicina, Universidade de Sao Paulo, Sao Paulo, SP, Brazil, 6 Departamento de Genética Médica e Medicina Genômica, Faculdade de Ciências Médicas, Universidade Estadual de Campinas, Campinas, São Paulo, Brazil, 7 Departamento de Pediatria, Faculdade de Ciências Médicas, Universidade Estadual de Campinas, Campinas, São Paulo, Brazil

¤ Current address: The University of Chicago, Department of Medicine, Section of Endocrinology, Chicago, IL, United States of America
* mmfrancabr@gmail.com

**Data Availability Statement:** All relevant data are within the manuscript and its Supporting Information files.

## Abstract

Primary ovarian insufficiency (POI) is a heterogeneous disorder associated with several genes. The majority of cases are still unsolved. Our aim was to identify the molecular diagnosis of a Brazilian cohort with POI. Genetic analysis was performed using a customized panel of targeted massively parallel sequencing (TMPS) and the candidate variants were confirmed by Sanger sequencing. Additional copy number variation (CNV) analysis of TMPS samples was performed by CONTRA. Fifty women with POI (29 primary amenorrhea and 21 secondary amenorrhea) of unknown molecular diagnosis were included in this study, which was conducted in a tertiary referral center of clinical endocrinology. A genetic defect was obtained in 70% women with POI using the customized TMPS panel. Twenty-four pathogenic variants and two CNVs were found in 48% of POI women. Of these variants, 16 genes were identified as *BMP8B*, *CPEB1*, *INSL3*, *MCM9*, *GDF9*, *UBR2*, *ATM*, *STAG3*, *BMP15*, *BMPR2*, *DAZL*, *PRDM1*, *FSHR*, *EIF4ENIF1*, *NOBOX*, and *GATA4*. Moreover, a microdeletion and microduplication in the *CPEB1* and *SYCE1* genes, respectively, were also identified in two distinct patients. The genetic analysis of eleven patients was classified as variants of uncertain clinical significance whereas this group of patients harbored at least two variants in different genes. Thirteen patients had benign or no rare variants, and therefore the genetic etiology remained unclear. In conclusion, next-generation sequencing (NGS) is a highly effective approach to identify the genetic diagnoses of heterogenous disorders, such as POI. A molecular etiology allowed us to improve the disease knowledge, guide decisions about prevention or treatment, and allow familial counseling avoiding future comorbidities.

**Funding:** The authors received the following funding in support of this study: Fundação de Amparo à Pesquisa do Estado de São Paulo, 2014/14231-0 to Dr. Monica M França; Fundação de Amparo à Pesquisa do Estado de São Paulo, 2013/02162-8 to Berenice B. Mendonca; Conselho Nacional de Desenvolvimento Científico e Tecnológico, 303002/2016-6 to Berenice B. Mendonca; and Fundação de Amparo à Pesquisa do Estado de São Paulo, 2014/50137-5.

**Competing interests:** The authors have declared that no competing interests exist.

## Introduction

Primary ovarian insufficiency (POI), also known as premature ovarian failure (POF), results in primary or secondary amenorrhea, hypoestrogenism, infertility, and elevated gonadotrophin levels (FSH>LH) [1]. POI patients have shown widely varying clinical phenotypes starting with women at puberty up to 40-year-old women. These patients can present primary amenorrhea, usually diagnosed at a younger age with delay of puberty and absence of breast development, whereas secondary amenorrhea is diagnosed at ages from <20 up to 40 years and is characterized by an irregular menstrual cycle and most often normal pubertal development and is the most frequent POI phenotype [2].

POI appears to have a genetic component presenting as sporadic or familial. However, the inheritance pattern is known to be autosomal dominant or autosomal recessive and monogenic, and recently, oligogenic inheritance has been proposed [3]. The heterogenous genetic basis of POI can rely on over 75 genes [2]. These genes are involved in several pathways such as gonadal development, meiosis (DNA replication and repair), hormonal signaling, immune function, and metabolism, although the majority of POI cases are yet to be elucidated [2,4]. This limitation may be narrowed with advancements in massively parallel sequencing, also known as next-generation sequencing (NGS). This method has been used as an effective tool to elucidate the genetic origin of heterogeneous diseases, such as POI. The aim of this study was to identify the genetic diagnosis of 50 patients with POI by using targeted massively parallel sequencing (TMPS).

## Patients and methods

**POI cohort.**   Approval from the institutional review board and written informed consent were obtained from all subjects or the parents or guardians of the minors before blood collection for DNA analysis. This study was approved by the Ethics Committee of Hospital das Clínicas, University of São Paulo Medical School, Brazil (protocol number 2015/12837/1.015.223). A cohort of 50 women presenting with POI was selected for this study between 2014 and 2018 at the Hospital das Clínicas University of São Paulo Medical School (Table 1). All patients had high levels of gonadotropins (FSH>20 U/L), hypoestrogenism, absence of *FMR1* premutation, and positive anti-ovary or 21-hydroxylase antibodies. In primary amenorrhea cases, the patients also presented delay of puberty and absence of spontaneous breast development. The patients were treated with conjugated estrogens daily followed by progesterone replacement in the first 12 days of the month, resulting in menstrual bleeding and complete breast development in patients with primary amenorrhea.

## Molecular analyses

**Targeted Massively Parallel Sequencing (TMPS).**   Genomic DNA was extracted from peripheral blood leukocytes using standard procedures. A custom SureSelect<sup>XT</sup> DNA target enrichment panel was designed using SureDesign tools (Agilent Technologies Santa Clara, CA, USA). Based on human and animal gonadal development and function, known and candidate genes were selected as follows:

1) *Gonadal formation*: CBX2, CTNNB1, DHH, EMX2, FGF9, FST, GATA4, LHX9, NR0B1, NR5A1, RSPO1, SOX9, SOX8, SRY, WNT4, WT1, and ZFPM2;

2) *Ovarian development*: AMH, AMHR2, BCL2, BCL2L2, BMP15, BMP4, BMP8B, BMPR1B, BMPR2, CDKN1B, CYP11A1, CYP17A1, CYP19A1, DAZL, DIAPH2, DND1, EIF2B2, EIF2B5, eIF4ENIF1, FIGLA, FMN2, FOXL2, FOXO1, FOXO3, FOXO4, FSHR, GDF9, GJA4, INHA, INHBA, INHBB, INSL3, KIT, KITLG, LHCGR, LHX8, NANOS1, NANOS2, NANOS3, NBN, NOBOX, PGRMC1, POF1B, POLR3H, POR, POU5F1, PDGFRA, PRDM1, PTEN, SMAD1,

**Table 1. Clinical features of 50 Brazilian women with primary ovarian insufficiency.**

| Patient ID | Amenorrhea | Age at first appointment (yr) | Previous treatment | FSH (IU/L) | LH (IU/L) | Height (cm) | Tanner stage at diagnosis | Syndromic features | Associate phenotype |
|---|---|---|---|---|---|---|---|---|---|
| POI-1 | Secondary | 27 | No | 76 | 43 | 163 | B5/P5 | | |
| POI-2 | Primary | 23 | Yes (at 17yr) | 99 | 45 | 166 | B5/P4 | High arched palate; Cubitus valgus | |
| POI-3 | Primary | 19 | | 29 | 9 | 170 | B1/P2 | Cubitus valgus | Hearing loss, sensorineural |
| POI-4 | Primary | 17 | | 128 | 51 | 147 | B1/P1 | | |
| POI-5 | Primary | 32 | Yes | 48 | 25 | 161 | B4/P4 | | Type 2 Diabetes mellitus |
| POI-6 | Primary | 17 | Yes (at 16yr) | 65 | 25 | 166 | B5/P4 | | |
| POI-7 | Primary | 17 | | 21 | 12 | NA | B1/P2 | | |
| POI-8 | Secondary | 28 | Yes (at 16yr) | 71 | 21 | 168 | B4/P4 | | |
| POI-9 | Primary | 18 | | 118 | 64 | 145 | B1/P2 | High arched palate; Cubitus valgus; wide-spaced nipples, growth deficit | Macrocytic anemia |
| POI-10 | Secondary | 32 | | 89 | 19 | NA | NA | | |
| POI-11 | Primary | 13 | | 100 | 44 | 141 | B3/P3 | Short stature | |
| POI-12 | Secondary | 37 | | 38 | 32 | 147 | B5/P5 | | |
| POI-13 | Primary | 23 | | 83 | 28 | NA | B2/P4 | | |
| POI-14 | Secondary | 21 | Yes (at 19yr) | 63 | 39 | 162 | B2/P2 | | |
| POI-15 | Secondary | NA | | 64 | 20 | NA | NA | | |
| POI-16 | Primary | 19 | | 87 | 51 | 151 | B2/P3 | | |
| POI-17 | Secondary | 25 | | 76 | 36 | NA | NA | | |
| POI-18 | Primary | 19 | Yes (at 16yr) | 46 | 28 | 163 | B3/P4 | High arched palate | Dyslipidemia |
| POI-19 | Primary | 30 | Yes (at 12yr) | 58 | 24 | 156 | B2/P5 | High arched palate | |
| POI-20 | Primary | 26 | Yes (at 25yr) | 138 | 47 | 167 | B4/P4 | Cubitus valgus | Tremor on right/dominant hand; no muscle atrophy |
| POI-21 | Primary | 21 | | 89 | 37 | 161 | B3/P3 | | |
| POI-22* | Primary | 21 | Yes (at 17yr) | 94 | 25 | 164 | B5/P5 | | Migraine |
| POI-23* | Primary | 17 | Yes (at 15yr) | 45 | 21 | 176 | B4/P5 | | Wilson's disease and keratoconus |
| POI-24 | Primary | 16 | No | 106 | 42 | 157 | B1/P3 | Cubitus valgus; Late psychomotor development | |
| POI-25 | Secondary | 21 | | 96 | 61 | NA | NA | | |
| POI-26 | Primary | 14 | No | 87 | 51 | 164 | B1/P1 | High arched palate | Chronic telogen effluvium, hypothryroidism, type 2 diabetes mellitus |
| POI-27 | Secondary | 17 | Yes (at 15yr) | 98 | 27 | 164 | NA | | Precocious puberty (no treated) |
| POI-28 | Primary | 18 | | 48 | 10 | 167 | B1/P3 | | |
| POI-29 | Primary | 14 | | 114 | 34 | 150 | B1/P1 | | |
| POI-30 | Primary | 31 | Yes | 86 | 32 | 170 | NA | | |
| POI-31 | Secondary | 37 | | 103 | 10 | 160 | B5/P5 | | |
| POI-32 | Secondary | 34 | Yes (at 18yr) | 47 | 15 | 164 | B5/P5 | | |
| POI-33 | Secondary | 32 | Yes (at 28yr) | 18 | 14 | 148 | B5/P5 | | |
| POI-34 | Primary | 34 | Yes (at 18yr) | 65 | 30 | 156 | NA | | |
| POI-35 | Primary | 22 | Yes (at 15yr) | 42 | 19 | 157 | B4/NA | High arched palate; sindactilia | |
| POI-36 | Secondary | 38 | Yes (at 27yr) | 63 | 36 | 155 | B5/P5 | | |
| POI-37 | Primary | 14 | No | 96 | 32 | 157 | B1/P4 | Cafe au lait spots; high arched palate; cubitus valgus; hyperdontia | Congenital heart murmur |

*(Continued)*

**Table 1.** (Continued)

| Patient ID | Amenorrhea | Age at first appointment (yr) | Previous treatment | FSH (IU/L) | LH (IU/L) | Height (cm) | Tanner stage at diagnosis | Syndromic features | Associate phenotype |
|---|---|---|---|---|---|---|---|---|---|
| POI-38 | Primary | 23 | Yes (at 19yr) | 69 | NA | 154 | B4/P5 | | |
| POI-39 | Secondary | 19 | Yes (at 16yr) | 72 | 44 | 166 | B5/P4 | High arched palate | |
| POI-40 | Secondary | 38 | Yes (at 27yr) | 52 | 35 | 155 | B5/P5 | | Type 2 Diabetes mellitus |
| POI-41 | Secondary | 34 | | 38 | 13 | 168 | B5/P5 | | |
| POI-42 | Secondary | 35 | | 50 | 21 | 172 | B5/P5 | | |
| POI-43 | Secondary | 35 | | 88 | 35 | 151 | B4/P3 | | |
| POI-44 | Secondary | 30 | | 94 | 25 | 160 | B5/P5 | | |
| POI-45 | Primary | 43 | Yes (at 33yr) | 75 | 28 | 153 | B4/P5 | | Hearing loss, sensorineural; Kidney transplant, congenital heart murmur |
| POI-46 | Secondary | 40 | Yes (at 39yr) | 32 | 7 | 162 | B5/P5 | | |
| POI-47 | Secondary | 27 | Yes | 75 | 56 | 172 | B5/P5 | High arched palate, low-set posteriorly rotated ears | |
| POI-48 | Primary | 17 | No | 119 | 36 | 175 | B1/P4 | | Familial Ectrodactyly |
| POI-49[#] | Primary | 18 | | 63 | 32 | 155 | B1/P2 | | |
| POI-50[#] | Primary | 16 | | 66 | 28 | 151 | B2/P4 | | |

NA: not available.

[*] Siblings of Family 1.

[#] Sibling of Family 2.

*SMAD4, SMAD5, SOHLH1, SOHLH2, SOX3, STAR, STRA8, TCF21, TGFBR3, TIAL1, UBE3A,* and *ZFX*;

3) *Meiosis and DNA repair genes*: *ATM, BRWD1, CDC25B, CDK2, CKS2, CPEB1, CYP26B1, DMC1, ERCC1, ERCC2, CBS-PGBD3, FANCA, FANCC, FANCG, FANCL, GJA4, GPR3, HFM1, HSF2, MCM8, MCM9, MEI1, MLH1, MLH3, MOS, MSH4, MSH5, NOS3, NUP107, PMS2, PSMC3IP, RAD51B, REC8, SGOL2, SMC1B, SPO11, STAG3, SYCE1, SYCP1, SYCP2, SYCP3, TOP3B, TRIP13, UBB,* and *UBR2*;

4) *Putative variants, detected by GWAS, and causative genes without a clear mechanism in ovary function*: *ADAMTS16, ADAMTS19, BRSK1, CHM, COL4A6, DACH2, DMRT1, ERS1, ESR2, FMR1, HARS2, HK3, HSD17B4, LARS2, NCOA1, NXF5, PAPPA, RSPO2, TSHB,* and *XPNPEP2*;

5) *Candidate and known genes of disorders/differences in sex development*: *AKR1C2, AKR1C4, AR, ARX, ATRX, CDH7, CYP21A2, DHCR7, DMRT1, DMRT2, HNF1B, HSD11B1, HSD17B3, HSD3B2, FGFR2, LHX1, MAMLD1, MAP3K1, SRD5A2, SOX3,* and *WWOX.*

Exonic regions and 25 base pairs of intronic flanking sequence of all genes were included. The proband's libraries were prepared according to the SureSelect[XT] Target Enrichment Protocol (Agilent Technologies Santa Clara, CA, USA). Deep sequencing of these amplicon libraries was performed on a NextSeq 500 next-generation sequencer (Illumina San Diego, CA, USA). Alignment of raw data and variant calling were performed following the steps described by França and collaborators [5]. The first criterion used to distinguish new variants from polymorphisms was filtered variants with a MAF<0.01 in 1000 Genomes, Exome Variant Server NHLBI GO Exome Sequencing Project (ESP), and Exome Aggregation Consortium (ExAC) databases. The variants were evaluated in the gnomAD database and in the results section; only gnomAD is shown since all public databases described above are included in it. Moreover, only missense, nonsense, and frameshift variants in coding regions and splice sites were

included. The pathogenicity predictions for new point mutations leading to aminoacid changes were evaluated on SIFT, Polyphen2, Mutation Taster, CADD, and GERP. The allelic variants were classified based on the American College of Medical Genetics and Genomics and the Association for Molecular Pathology (ACMG/AMP) guidelines [6] combined with Inter-Var analysis [7], and additional information obtained from animal model findings already reported in the literature. For recessive inheritance (homozygous or compound heterozygous variants), a PM3 criterion was used. Three out of five computational in silico tools were used as evidence to support a deleterious effect on the gene or gene product (S1 Table). New mutations reported in this study were not validated at RNA and/or protein levels.

## Sanger sequencing

The mutations detected by TMPS were confirmed in the patients and their available families by Sanger sequencing. Primers flanking the variants were used for PCR amplification. Primer sequences are available on request. All PCR products were sequenced using the BigDye terminator v3.1 cycle sequencing kit followed by automated sequencing using the ABI PRISM 3130xl (Applied Biosystems, Foster City, CA). Moreover, Sanger sequencing was performed to evaluate 200 fertile Brazilian women controls for the putative damaging mutation found in the patients with *GATA4*, *GDF9*, *and STAG3*.

## Copy number variation

For analyzing the copy number variation (CNV) in TMPS samples, CONTRA (COpy Number Targeted Resequencing Analysis) software was utilized. This method for the detection of CNV using NGS data is based on empirical relationships between log-ratio and coverage and therefore is capable of identifying copy numbers of gains and losses for each target region based on normalized depth of coverage [8]. We evaluated log-ratio +1 for duplication and log-ratio -1 for deletion, adjusting the P-value below 0.05. Integrative Genomics Viewer (IGV) software was used to confirm the decreased or increased depth coverage [9,10]. Some CNVs were confirmed by Multiplex Ligation-dependent Probe Amplification (MLPA) (MRC Holland, Amsterdam, The Netherlands), when commercial probes were available. The MLPA reaction was performed according to the manufacturer's recommendations.

## Results

The mean coverage depth of the targeted regions data was x173.6 (SD ± ×79), with at least 98% of the sequenced bases covering more than 10-fold. We recruited 50 patients from different Brazilian endocrinology institutes, including 29 patients with primary amenorrhea and 21 with secondary amenorrhea (Table 1), all of them with the 46,XX karyotype. As listed in Table 1, 37 patients presented with isolated POI, and 13 syndromic POI, which were characterized by the presence of high arched palate, cubitus valgus, late psychomotor development, short stature, and other skeletal abnormalities.

A genetic defect was identified in 70% (35 of 50) of women with POI using the customized TMPS panel. A total of 24 pathogenic variants and 2 CNVs were identified in 48% (24 of 50) of POI patients and considered a molecular genetic diagnosis of POI. These twenty-four pathogenic variants are related to 16 genes: *BMP8B* (POI-1 and POI-36), *CPEB1* (POI-4), *INSL3* (POI-5), *MCM9* (POI-11 and POI-25), *GDF9* (POI-16, POI-49, and POI-50), *UBR2* (POI-17), *ATM* (POI-20), *STAG3* (POI-21), *BMP15* (POI-22 and POI-23), *BMPR2* (POI-29), *DAZL* (POI-31), *PRDM1* (POI-37), *FSHR* (POI-38), *EIF4ENIF1* (POI-40), *NOBOX* (POI-42), and *GATA4* (POI-12 and POI-45) (Table 2). These changes included 18 missense variants, 3 nonsense variants, and 3 frameshift variants. A total of 13 patients carried a single heterozygous

Table 2. Pathogenic variants detected in a Brazilian cohort of 50 women with primary ovarian insufficiency.

| Patient ID | Gene | Accession number | Genotype | Variant annotation | gnomAD[1] | dbSNP | Novel or previously reported variant in POI patient | ACMG classification[2] | Supporting evidence related to infertility/POI in animal models and humans |
|---|---|---|---|---|---|---|---|---|---|
| POI-1 | BMP8B | NM_001720.5 | Heterozygous | c.1024A>G: p.M342V | 0.00006 | rs149276444 | Novel | P: PS3+PM1 +PM2+PP2 +PP3 | Ref. [11] |
| POI-4 | CPEB1 | NM_030594.5 | Heterozygous | c.259C>T:p. R87C | 0.0004 | rs200188266 | Novel | LP: PS3+PM1 +PM2+PP3 +BP1 | Ref. [12–15] |
| POI-5 | INSL3 | NM_001265587.2 | Homozygous | c.52G>A:p. V18M | 0.001 | rs200056709 | Novel | LP: PS3+PM1 +PM3+PP2 +BP4 | Ref. [16] |
| POI-7 | Chr10: SYCE1 | NM_130784 | Heterozygous | 11.4Kb duplication | NA | NA | Novel | NA | Ref. [12–15,17] |
| POI-11 | MCM9 | NM_017696.2 | Compound Heterozygous | c.2059T>C: p.F687L | Absent | rs1046135510 | Novel | LP: PS3+PM2 +PM3+BP4 | Ref. [2,4,18–22] |
| | MCM9 | NM_017696.2 | Compound Heterozygous | c.3223C>T: p.P1075S | 0.003 | rs61744508 | Novel | LP: PS3+PM3 +BP4 | Ref. [2,4,18–22] |
| POI-12 | GATA4 | NM_002052.5 | Heterozygous | c.280G>A:p. A94T | 0.0001 | rs780764610 | Novel | LP: PS3+PM1 +PM2+PP2 +BP4 | Ref. [23] |
| POI-14 | Chr15: CPEB1 | NM_030594 | Heterozygous | 83.8Kb deletion | NA | NA | Novel | NA | Ref. [12–15] |
| POI-16 | GDF9[3] | NM_005260.5 | Homozygous | c.783delC:p. S262Hfs*2 | Absent | rs1216260561 | Novel | P: PVS1+PS3 +PM1 +PM2PM3+PP3 | Ref. [2,4,19,22,24] |
| POI-17 | UBR2 | NM_015255.2 | Heterozygous | c.4843T>A: p.S1615T | Absent | rs1017000245 | Novel | LP: PS3+PM2 +PP2+PP3 | Ref. [25] |
| POI-20 | ATM | NM_000051.3 | Heterozygous | c.334G>A:p. A112T | 0.0002 | rs146382972 | Novel | LP: PM1+PM2 +PM3+PP3 +PP5+BP1 | Ref. [4] |
| | ATM | NM_000051.3 | Heterozygous | c.7313C>T: p.T2438I | 0.0001 | rs147604227 | Novel | LP: PM1+PM2 +PM3+PP3 +PP5+BP1 | Ref. [4] |
| POI-21 | STAG3[4] | NM_001282716.1 | Heterozygous | c.290dupC:p. N98Qfs*2 | Absent | Absent | Novel | P: PVS1+PS3 +PM2+PM3 +PP3+PP5 | Ref. [2,4,19,22,26,27] |
| | STAG3[4] | NM_001282716.1 | Heterozygous | c.1950C>A: p.Y650* | Absent | Absent | Novel | P: PVS1+PS3 +PM2+PM3 +PP3 | Ref. [2,4,19,22,26,27] |
| POI-22 | BMP15* | NM_005448.2 | Homozygous | c.343C>T:p. Q115* | 0.00001 | rs782799707 | Novel | P: PVS1+PS3 +PM1+PM2 +PP3 | Ref. [4,19,22,28,29] |
| POI-23 | BMP15* | NM_005448.2 | Homozygous | c.343C>T:p. Q115* | 0.00001 | rs782799707 | Novel | P: PVS1+PS3 +PM1+PM2 +PP3 | Ref. [4,19,22,28,29] |
| POI-25 | MCM9 | NM_017696.2 | Heterozygous | c.1163C>A: p.T388N | 0.0005 | rs545524695 | Novel | LP: PS3+PM1 +PM2+PP3 +BP1 | Ref. [18,20,21] |
| POI-29 | BMPR2 | NM_001204.7 | Heterozygous | c.1357G>A: p.V453M | Absent | Absent | Novel | P: PS3+PM1 +PM2+PP2 +PP3 | Ref. [30] |
| POI-31 | DAZL | NM_001190811.1 | Heterozygous | c.640C>T:p. Q214* | Absent | Absent | Novel | P: PVS1+PM2 +PP3 | Ref. [31,32] |

(Continued)

**Table 2.** (Continued)

| Patient ID | Gene | Accession number | Genotype | Variant annotation | gnomAD[1] | dbSNP | Novel or previously reported variant in POI patient | ACMG classification[2] | Supporting evidence related to infertility/POI in animal models and humans |
|---|---|---|---|---|---|---|---|---|---|
| POI-36 | *BMP8B* | NM_001720.5 | Heterozygous | c.778C>T:p. R260C | 0.002 | rs199806017 | Novel | LP: PS3+PM1 +PP2+PP3 | Ref. [11] |
| POI-37 | *PRDM1* | NM_001198.4 | Heterozygous | c.1250C>G: p.P417R | Absent | rs200035233 | Novel | LP: PS3+PM2 +PP2+PP3 | Ref. [33,34] |
| POI-38 | *FSHR* | NM_000145.4 | Compound Heterozygous | c.1298C>A: p.A433D | 0.000008 | rs763676828 | Reported (Ref. [5]) | P: PS3+PM1 +PM2+PM3 +PP2+PP3+PP5 | Ref. [2,4,5,22] |
| | *FSHR* | NM_000145.4 | Compound Heterozygous | c.507delC:p. F170Lfs*4 | 0.000004 | rs746673169 | Novel | P: PVS1+PS3 +PM2+PM3 +PP3 | Ref. [2,4,22] |
| POI-40 | *EIF4ENIF1* | NM_001164501.2 | Heterozygous | c.2006A>G: p.K669R | 0.00002 | rs374538489 | Novel | LP: PS3+PM2 +PP3 | Ref. [35–37] |
| POI-42 | *NOBOX* | NM_001080413.3 | Heterozygous | c.479C>T:p. P160L | 0.00003 | rs372037920 | Novel | LP: PS3+PM1 +PP2+BP4 | Ref. [2,4,38,39] |
| POI-45 | *GATA4* | NM_002052.5 | Homozygous | c.1220C>G: p.P407R | 0.00006 | rs115099192 | Novel | P: PS3+PM2 +PM3+PM5 +PP2+PP3+BP1 | Ref. [23] |
| POI-49 | *GDF9*[#] | NM_005260.5 | Heterozygous | c.389A>G:p. Q130R | Absent | Absent | Novel | LP: PS3+PM2 +PP1+PP2+BP4 | Ref. [2,4,19,22,24] |
| POI-50 | *GDF9*[#] | NM_005260.5 | Heterozygous | c.389A>G:p. Q130R | Absent | Absent | Novel | LP: PS3+PM2 +PP1+PP2+BP4 | Ref. [2,4,19,22,24] |

[*] Siblings of Family 1

[#] Siblings of Family 2.

[1] The variant frequency was assessed in the gnomAD database (https://gnomad.broadinstitute.org/). Accessed in July 2019.

[2] ACMG/AMP classification was done according to Richards et al. (Ref. [6]) combined with InterVar evaluation (Ref. [7]). Accessed in June 2020.

[3] Published as a case report in Franca et al., 2018 (Ref. [24]).

[4] Published as a case report in Franca et al., 2019 (Ref. [26]).

P: Pathogenic variant; LP: Likely Pathogenic variant; VUS: Variant of uncertain significance; LB: Likely Benign variant; B: Benign variant. NA: not available.

pathogenic variant (13 of 22, 59%), 4 patients presented compound heterozygous variants (4 of 22, 18%), and 5 homozygous variants were found (5 of 22, 23%). Furthermore, two pathogenic CNVs were detected in 2 patients (2 of 50, 4%). Among the identified CNVs, POI-14 had a microdeletion in the *CPEB1* gene (83.8-Kb) and POI-7 carried a microduplication of the *SYCE1* gene (11.4-Kb) (Table 2).

In this current study, eleven patients (11 of 50, 22%) harbored more than one variant in different genes (Table 3), and most of these variants were classified as variants of uncertain clinical significance (VUS). A total of 24 variants were identified in 16 unrelated genes (*POU5F1*, *HK3*, *NXF5*, *GATA4*, *NBN*, *ATM*, *COL4A6*, *XPNPEP2*, *SYCP1*, *FANCL*, *ERCC2 NOBOX*, *HARS2*, *SMC1B*, *GDF9*, *HELQ*).

Among the identified VUS, 19 variants were heterozygous and 5 were homozygous. Most of the identified VUS variants were missense (21 of 24, 88%), three were frameshifts (2 of 24, 8%), and one was a nonsense variant (1 of 24, 4%). Therefore, the first reported variants in novel genes and/or mode of inheritance in POI patients supported by animal findings are listed in Table 4.

In addition, 12 of 50 patients (24%) had no rare variants of these screened genes. Two potential pathogenic missense variants in the *BMP8B and ATM* genes and one 14.4 Kb microdeletion in the *TOP3B* gene were identified in 3 patients (POI-8, POI-32, and POI-48).

**Table 3. Variants detected in multiple genes in a Brazilian cohort of 50 women with primary ovarian insufficiency.**

| Patient ID | Gene | Accession number | Genotype | Variant annotation | gnomAD[1] | dbSNP | Novel or previously reported variant in POI patient | ACMG classification[2] | Supporting evidence related to infertility/ POI in animal models and humans |
|---|---|---|---|---|---|---|---|---|---|
| POI-3 | POU5F1 | NM_002701.6 | Heterozygous | c.133C>T:p.P45S | Absent | Absent | Novel | VUS: PM2+BP4 | Ref. [19] |
| | HK3 | NM_002115.3 | Heterozygous | c.1945C>T:p.R649C | 0.00005 | rs376092049 | Novel | VUS: PM1+PM2+PP3+BP1 | Ref. [19] |
| POI-9 | NXF5 | NM_032946.2 | Homozygous | c.959G>A:p.R320Q | 0.0004 | rs113591248 | Novel | LP: PM1+PM2+PM3+BP4 | Ref. [19,40] |
| | NXF5 | NM_032946.2 | Homozygous | c.145A>G:p.I49V | 0.0003 | rs113468014 | Novel | LP: PM1+PM2+PM3+BP4 | Ref. [19,40] |
| | GATA4 | NM_002052.5 | Heterozygous | c.280G>A:p.A94T | 0.0001 | rs780764610 | Novel | LP: PS3+PM1+PM2+PP2+BP4 | Ref. [23] |
| | NBN | NM_002485.5 | Heterozygous | c.456G>A:p.M152I | 0.0001 | rs201816949 | Novel | VUS: PM1+PM2+PP3+BP1 | Ref. [4] |
| POI-9 | ATM | NM_000051.3 | Heterozygous | c.5879T>A:p.I1960N | 0.000004 | rs587782503 | Novel | VUS: PM1+PM2+PP3+BP1 | Ref. [4] |
| POI-10 | NXF5 | NM_032946.2 | Heterozygous | c.958C>T:p.R320* | 0.001 | rs140252282 | Novel | VUS: PVS1+PP3+BP6 | Ref. [19,40] |
| | COL4A6 | NM_033641.4 | Heterozygous | c.2371G>A:p.G791S | 0.001 | rs143895379 | Novel | VUS: PP3+BP6 | Ref. [19] |
| | XPNPEP2 | NM_003399.6 | Heterozygous | c.644C>T:p.T215I | 0.002 | rs138365897 | Novel | VUS: PP3 | Ref. [19] |
| POI-13 | NXF5 | NM_032946.2 | Heterozygous | c.526G>A:p.G176S | 0.000006 | Absent | Novel | VUS: PM1+PM2+PP3 | Ref. [19,40] |
| | SYCP1 | NM_003176.4 | Heterozygous | c.433C>G:p.R145G | 0.000004 | Absent | Novel | VUS: PM2+PP3 | Ref. [41] |
| POI-15 | POU5F1 | NM_002701.6 | Heterozygous | c.87G>T:p.W29C | 0.0001 | rs200769740 | Novel | VUS: PM2+PP3 | Ref. [19] |
| | HK3 | NM_002115.3 | Heterozygous | c.2026C>T:p.P676S | 0.0003 | rs199684264 | Novel | VUS: PM2+PP3+BP1 | Ref. [19] |
| POI-24 | FANCL | NM_001114636.1 | Homozygous | c.1111_1114dup:p.T372Nfs*13 | 0.003 | rs759217526 | Novel | P: PVS1+PM3+PP3+BS2 | Ref. [42] |
| | ATM | NM_000051.3 | Homozygous | c.1273G>T:p.A425S | 0.000004 | rs769214234 | Novel | VUS: PM1+PM2+PM3+PM5+BP1+BP4 | Ref. [4] |
| POI-30 | ERCC2 | NM_000400.4 | Heterozygous | c.1606G>A:p.V536M | 0.0002 | rs142568756 | Novel | VUS: PM1+PM2+PP3 | Ref. [43] |
| POI-35 | NOBOX | NM_001080413.3 | Heterozygous | c.271G>T:p.G91W | 0.003 | rs77587352 | Reported (Ref. [39]) | LB: PS3+PP5+BS2+BP4 | Ref. [2,4,38,39] |
| | HK3 | NM_002115.3 | Heterozygous | c.521C>T:p.T174M | 0.002 | rs141123858 | Novel | VUS: PM1+PP3+BP1 | Ref. [19] |
| POI-41 | HARS2 | NM_012208.4 | Heterozygous | c.1105G>C:p.G369R | 0.0007 | rs61736946 | Novel | VUS: PM1+PP3+PP5 | Ref. [22] |
| | SMC1B | NM_148674.5 | Heterozygous | c.2683C>T:p.R895W | 0.0001 | rs199797179 | Novel | VUS: PM1+PP3+BP1 | Ref. [22,44] |
| POI-43 | SYCP1 | NM_003176.4 | Heterozygous | c.1747C>G:p.L583V | 0.0001 | rs147626229 | Novel | VUS: PM2+PP3 | Ref. [41] |
| POI-44 | GDF9 | NM_005260.5 | Heterozygous | c.191C>T:p.A64V | 0.00004 | rs751002918 | Novel | P: PS3+PM2+PP3 | Ref. [2,4,19,22,24] |
| | FANCL | NM_001114636.1 | Heterozygous | c.1111_1114dup:p.T372Nfs*13 | 0.003 | rs759217526 | Novel | VUS: PVS1+PP3+BS2 | Ref. [42] |

(*Continued*)

**Table 3.** (Continued)

| Patient ID | Gene | Accession number | Genotype | Variant annotation | gnomAD[1] | dbSNP | Novel or previously reported variant in POI patient | ACMG classification[2] | Supporting evidence related to infertility/ POI in animal models and humans |
|---|---|---|---|---|---|---|---|---|---|
| | *HELQ* | NM_133636.4 | Heterozygous | c.3095delA:p. Y1032Sfs*4 | 0.00002 | rs761786816 | Novel | P: PVS1+PM2 +PP3 | Ref. [45] |

* Siblings of Family 1

# Siblings of Family 2.

[1]The variant frequency was assessed in the gnomAD database (https://gnomad.broadinstitute.org/). Accessed in July 2019.

[2]ACMG/AMP classification was done according to Richards et al. (Ref. [6]) combined with InterVar evaluation (Ref. [7]). Accessed in June 2020.

P: Pathogenic variant; LP: Likely Pathogenic variant; VUS: Variant of uncertain significance; LB: Likely Benign variant; B: Benign variant.

However, these variants/CNVs were not considered to be causative since the fertile mothers also carried these deletion/variants, and thus they were classified as benign. Thereby, fifteen POI patients have remained without a genetic diagnosis.

## Discussion

POI is a heterogeneous disorder characterized by a strong genetic basis that comprises at least 75 genes [2]. Defects in genes involved in gonadal development (oogenesis and folliculogenesis), meiosis and DNA repair, hormonal signaling, immune function, and metabolism are related to the POI phenotype [2,4]. Herein, a genetic defect of POI patients was obtained in 70% of affected women using a customized TMPS panel (Tables 2 and 3) and is discussed below. The majority of these defects were found in autosomal genes related to oogenesis and folliculogenesis and meiosis/DNA repair genes summarized in Fig 1.

**Table 4. List of novel genes or mode of inheritance in primary ovarian insufficiency patients supported by animal model findings.**

| Patient ID | Gene | Genotype | Variant annotation | Supported by animal model findings | Inheritance or genotype previously identified | References |
|---|---|---|---|---|---|---|
| POI-1 | *BMP8B* | Heterozygous | c.1024A>G:p.M342V | Reduced number of PGCs | - | [11] |
| POI-36 | *BMP8B* | Heterozygous | c.778C>T:p.R260C | Reduced number of PGCs | - | [11] |
| POI-5 | *INSL3* | Homozygous | c.52G>A:p.V18M | Disruption of female cycle and reduced number of litters | - | [16] |
| POI-12 | *GATA4* | Heterozygous | c.280G>A:p.A94T | Regulation of ovarian steroidogenesis | - | [23] |
| POI-45 | *GATA4* | Heterozygous | c.1220C>G:p.P407R | Regulation of ovarian steroidogenesis | - | [23] |
| POI-16 | *GDF9*[3] | Homozygous | c.783delC:p.S262Hfs*2 | Block in follicular development leading to complete infertility | Heterozygous/Missense | [2,4,46] |
| POI-17 | *UBR2* | Heterozygous | c.4843T>A:p.S1615T | Reduced fertility | - | [25] |
| POI-30 | *ERCC2* | Heterozygous | c.1606G>A:p.V536M | No signs of estrus cycle, small ovaries, and no preovulatory follicles | - | [43] |
| POI-31 | *DAZL* | Heterozygous | c.640C>T:p.Q214* | Subfertility | Missense | [31,32] |
| POI-37 | *PRDM1* | Heterozygous | c.1250C>G:p.P417R | Arterial pole defects, reduced and failed migration and proliferation of PGCs | - | [33,34] |
| POI-44 | *FANCL* | Homozygous and Heterozygous | c.1111_1114dup:p. T372Nfs*13 | Reduced fertility and defective of germ cells | - | [42] |
| POI-44 | *HELQ* | Heterozygous | c.3095delA:p. Y1032Sfs*4 | Subfertility and germ cell attrition | - | [45] |

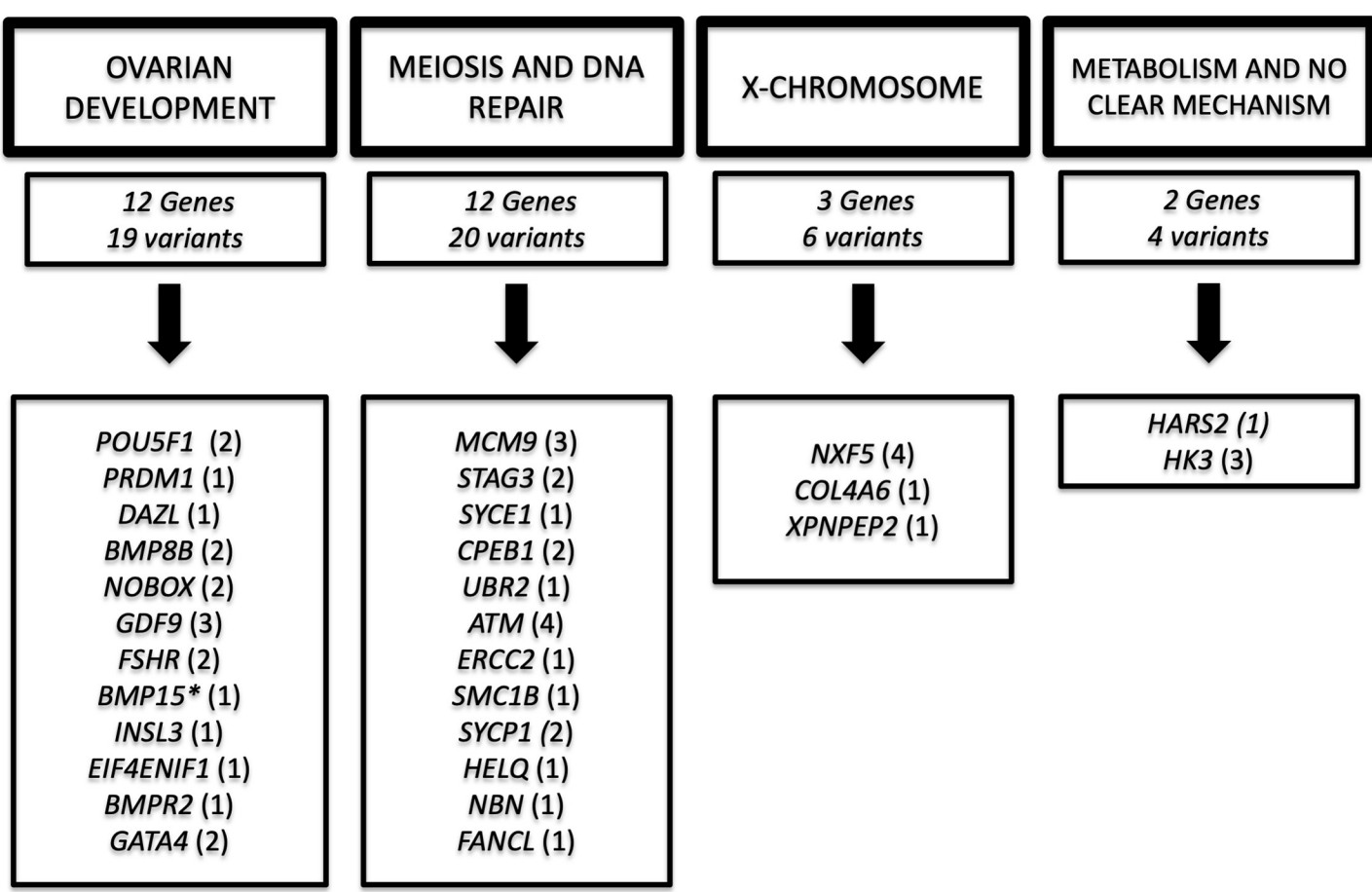

**Fig 1. Diagram showing the number of variants classified as pathogenic, likely pathogenic, and variant of uncertain significance in each gene group identified in 50 primary ovarian insufficiency patients.** *BMP15* is located on the X-chromosome; however, it is shown in ovarian development due to its well-known role in this category.

## A) Ovarian development: Oogenesis and foliculogenesis

Genes related to primordial germ cells (*POU5F1*, *PRMD1*, *DAZL*, *BMP8B)*

Primordial germ cells (PGCs) emerge from the extraembryonic mesoderm and migrate to the genital ridge, giving rise to oocytes [22]. Several proteins are involved in migration, proliferation, and survival of PGCs, such as TGFB factors and Wnt/B-catenin pathways. PRDM1 is able to drive these pathways by obtaining pluripotent cells [22]. Moreover, POU5F1 (OCT4), DAZL, and SYCP1 are also implicated in oocyte development.

**POU5F1.** POU class 5 homeobox 1 is a pluripotent gene downregulated in *Nobox KO* mice [19]. One heterozygous missense variant (p.Pro13Thr) in *POU5F1* was reported in one Chinese POI woman [19], and herein, one novel heterozygous variant in *POU5F1* (c.133C>T; p.Pro45Ser), combined with a second novel heterozygous variant in the *HK3* gene (Table 3), was found in a 19-year-old woman presenting with secondary amenorrhea, cubitus valgus, and hearing loss (POI-3) (Table 1). We also identified a second combination of *POU5F1* (c.87G>T;p.Trp29Cys) and HK3 (c.2026C>T;p.Pro676Ser) (Table 3) in POI-15 presenting with secondary amenorrhea (Table 1). Interestingly, *HK3* was revealed as a potential candidate leading to POI previously identified by GWAS associated with susceptibility between POI and early menopause [19].

**PRMD1.** One novel heterozygous missense variant in *PRDM1* (Table 2) was found in POI-37, who was diagnosed at 14 years of age presenting with primary amenorrhea, delay of puberty, cafe au lait spots, high arched palate, cubitus valgus, hyperdontia, and congenital heart defect (Table 1). Consistent with this associated phenotype, *Prdm1* mutation in mouse embryos promoted arterial pole defects characterized by misalignment or reduction of the aorta and pulmonary trunk and abnormalities in the arterial tree [33]. Moreover, heterozygous and null *Prdm1*-deficient mutant embryos form a tight cluster of PGCs that fail to show the migration, proliferation and repression of homeobox genes (Hox) [34].

**DAZL.** DAZL expression (Deleted in azoospermia-like), specifically expressed in germ cells, is essential in the beginning of meiosis, as it induces STRA8 and, consequently, the activation of SYCP1, 2 and 3, which are members of the synaptonemal complex [22]. Jung and collaborators [31] have shown by expressing DAZL with recombinant human GDF9 and BMP15 that embryonic stem cells can be induced to differentiate into ovarian follicle-like cells, such as oocytes and granulosa cells, underlying DAZL as a key player in ovarian development. In addition, the homozygous female and male *Dazl*[-/-] mice are infertile, while heterozygous mice exhibit a subfertile phenotype. The novel nonsense heterozygous variant (c.640C>T;p. Gln214*) in *DAZL* (Table 2) was identified in POI-31, who was diagnosed with secondary amenorrhea at 37 years of age (Table 1). Heterozygous and homozygous missense variants in *DAZL* were reported in infertile men and woman associated with secondary amenorrhea [32]. Although *DAZL* plays a key role in human and animal germ cells, it seems that pathogenic variants are a rare cause of male and female infertility.

**SYCP1.** Although SYCP1 is an essential member of the synaptonemal complex in labeling the axes of the chromosome during meiotic prophase I [22], no pathogenic variants associated with female infertility have been identified in this gene. *Sycp1*[-/-] male and female mice were infertile, whereas *Sycp1*[+/-] mice were fertile [41]. Herein, POI-43, who presented secondary amenorrhea at 35 years of age (Table 1), has a novel heterozygous missense variant in *SYCP1* (c.1747C>G;p.Leu583Val), which is rare and deleterious according to *in silico* analysis (Table 3). Further investigation of heterozygous variants in *SYCP1* should be done in order to evaluate a dominant negative or haploinsufficiency effect in this patient.

**BMP8B.** BMP8B, a member of the BMP superfamily, is also implicated in the primordial germ cell (PGC) process through the events of the development stages to generate mature sperm and oocytes. *Bmp8b*[-/-] male mice have shown small testes and infertility [47]. In addition, *Bmp8b* is required for PGC generation in female rodent physiology, whereas null and heterozygous *Bmp8b* mice have shown a lack of PGCs and reduced number of PGCs, respectively. In this study, two patients had novel and deleterious heterozygous *BMP8B* variants (POI-1: c.1024A>G;p.Met342Val; POI-36:c.778C>T;p.Arg260Cys) (Tables 2 and 4). POI-1 and POI-36 were diagnosed with secondary amenorrhea, and no additional phenotype or syndromic features were found (Table 1). Although no mutations in the *BMP8B* gene have been reported in POI patients, the decreased number of PGCs described in rodents could explain the secondary amenorrhea phenotype, and these mutations could lead to POI in these patients [11].

## Genes related to folliculogenesis (*NOBOX, GDF9, FSHR, BMP15, INSL3, EIF4ENIF1, BMPR2, GATA4*)

**NOBOX.** *NOBOX, GDF9, and FSHR* are well-known genes associated with POI, and a summary of these genes was described in França et al [5,24,38]. Here, POI-42, who was diagnosed at 35 years of age presenting with secondary amenorrhea (Table 1), harbored a novel and pathogenic heterozygous variant in *NOBOX* (c.479C>T:p.Pro160Leu) (Table 2). Moreover, POI-35 has a novel heterozygous missense variant in *HK3* (Table 3), which is associated

with an early age of menopause [19], and she also has a reported *NOBOX* variant (c.271G>T: p.Gly91Trp) (Table 3) previously described by Bouilly and collaborators [39]. A cosegregation was evaluated in the unaffected mother of POI-35, and both variants were also found in her. This *NOBOX* variant has been classified as a likely benign variant according to ACMG guidelines due to its frequency in the population database (0.003), absence of segregation, and no impact on the gene according to *in silico* analysis. However, Bouilly and collaborators [39] demonstrated impaired transcriptional activity of *NOBOX*_p.Gly91Trp for binding to the *GDF9* promoter *in vitro*. Hence, we classified this variant as VUS. These authors described two patients harboring this variant: one woman presenting with primary amenorrhea and absence of puberty without segregation data and another woman presenting with secondary amenorrhea and cosegregation of this variant in her family. Indeed, they were unable to rule out additional factors to explain the different phenotypes in those patients, and we were also unable to determine the genetic etiology of POI-35 since the fertile mother carried both variants. Further investigation should be performed on this patient.

**FSHR.** A pathogenic homozygous missense variant in *FSHR* was identified in two siblings presenting with hypergonadotropic hypogonadism using whole-exome sequencing [5] and references therein. In this study, one novel heterozygous frameshift deletion (c.507delC:p. Phe170Leufs*4) and one reported heterozygous missense (c.1298C>A:p.Ala433Asp) in the *FSHR* (Table 2) were identified in POI-38, a 23-year-old girl presenting with primary amenorrhea and delayed puberty (Tables 1 and 2). Interestingly, the c.1298C>A:p.Ala433Asp homozygous missense variant in the *FSHR* gene was previously reported by our group as causative of POI [5].

**GDF9.** The first homozygous 1-bp deletion in the *GDF9* gene was identified in POI-16 with primary amenorrhea published as a case report [24]. Furthermore, a novel pathogenic heterozygous missense (c.389A>G:p.Gln130Arg) was found in POI-49 and her affected sister (POI-50) (Table 2). Both siblings were diagnosed with primary amenorrhea, puberty delay, and no other associate features were found (Table 1). Interestingly, POI-44 carried a rare and novel heterozygous missense variant in *GDF9* (c.191C>T;p.Ala64Val), but two other variants in distinct genes were also found to cause a frameshift insertion in *FANCL* and a novel frameshift deletion in *HELQ* (c.3095delA;p.Tyr1032Serfs*4) (Table 3). Although the *FANCL* gene has been associated with Fanconi anemia, no clinical features of Fanconi were found in this patient, and therefore it is unlikely that the *FANCL* gene might cause ovarian failure. HELQ is a member of the DNA repair process related to tumor predisposition. In addition, *Helq*-deficient female mice showed subfertility and germ cell attrition. Moreover, heterozygous female mice exhibited a similar but less severe phenotype, indicative of haploinsufficiency [45] correlating to secondary amenorrhea found in POI-44 (Table 1). A cohort of Chinese women with POI was evaluated by Sanger and no mutations were found in the *HELQ* gene [48]. Unfortunately, the parental DNAs were unavailable for segregation analysis. Therefore, the molecular diagnosis of POI-44 needs further effort and remains uncertain, as we were unable to rule out a combined effect of these variants.

**BMP15.** Bone morphogenetic protein 15 has already been reported as an X-linked POI cause in patients with primary and secondary amenorrhea (MIM 300510), and recently, a homozygous missense variant in this gene was described in a patient with secondary amenorrhea [28]. In addition, homozygous ablation of BMP15 in sheep caused impaired follicular development beyond the primary stage [29]. In contrast with these models, $Bmp15^{-/-}$ mice exhibited minimal ovarian histopathological defects but showed reduced ovulation and fertilization rates and no phenotype was observed in $Bmp15^{+/-}$ mice. In this study, a nonsense homozygous variant (c.343C>T;p.Gln115*) in the *BMP15* gene was identified in two siblings (POI-22 and POI-23) with primary amenorrhea (Table 2). Both affected sisters presented

delayed puberty with hypergonadotropic hypogonadism (Table 1). Although heterozygous pathogenic variants in *BMP15* have been reported in POI patients, the mechanism of haploin-sufficiency or dominant negative effect was barely demonstrated [22]. Indeed, this is the first familial case showing a fertile mother carrier of a pathogenic heterozygous variant, arguing in favor of no heterozygous phenotype being found in animal models.

**INSL3.** INSL3 is a member of the insulin-like group of peptide hormones. It was first identified as a testis-specific gene transcript sequence in Leydig cells. Nevertheless, INSL3 is also produced in steroidogenic theca internal cells of antral follicles, which are equivalent cells to Leydig cells in the female physiology in bovines, rodents, monkeys, and humans [16]. INSL3 plays a key role in androstenedione synthesis, a major steroid precursor for granulosa cells to produce estrogens. In fact, the loss of INSL3 was shown to be a marker for follicle atre-sia, earlier than any steroidogenic expression depletion. Furthermore, the role of *INSL3* in POI pathology is poorly understood since no patients have been described yet. In this study, POI-5, born from a consanguineous marriage, was evaluated at 32 years of age with a history of pri-mary amenorrhea (Table 1). She harbored a novel homozygous variant in the exon 1 of the *INSL3* gene (c.52G>A:p.Val18Met) (Table 2). In addition, knockout female mice have shown an impairment in fertility with disruption of the female cycle and reduced number of litters [16].

**EIF4ENIF1.** Eukaryotic translation initiation factor 4E nuclear import factor 1 (EIF4E-NIF1) has been implicated in female germ cell development as a translational repressor in fruit fly, rodents, and human [35,36]. Functional studies in *Drosophila* and mice demonstrated that haploinsufficiency of EIF4ENIF1 promotes abnormal oocyte growth and impaired meiotic maturation [35,36]. In addition, a novel heterozygous nonsense variant (p.Ser429*) in *EIF4E-NIF1* was found in one woman with isolated POI and secondary amenorrhea [37]. A rare and deleterious heterozygous missense variant (c.2006A>G:p.Lys669Arg) in this gene (Table 2) was identified in POI-40, who was also diagnosed with secondary amenorrhea and born from a nonconsanguineous marriage (Table 1). This second report reinforces the *EIF4ENIF1* gene as a dominant inheritance of POI.

**BMPR2.** Bone morphogenetic protein receptor type II is a transmembrane serine/threo-nine kinase that belongs to the TGF-beta superfamily, of which GDF9 and BMP15 are also members. In addition, BMPR2, the main genetic cause of pulmonary hypertension, is involved in embryonal development and in bone formation [49]. A POI patient presenting with an iso-lated secondary amenorrhea phenotype was reported to be harboring one heterozygous variant (p.Ser987Phe) in *BMPR2* in association with a second heterozygous variant in *LHCGR* (p. Ans99Ser) [50]. No functional studies were exhibited regarding the oligogenic hypothesis, although *in vitro* results of the p.Ser987Phe in *BMPR2* showed a potential implication of this gene in POI [30]. In this study, POI-29 was diagnosed at 14 years of age with primary amenor-rhea, puberty delay, and normal height (Table 1). She has a novel heterozygous variant in *BMPR2* (c.1357G>A;p.Val453Met) (Table 2). Consistent with the first report [50], no clinical features of pulmonary hypertension have been found in POI-29, but we cannot exclude future development of this disorder.

**GATA4.** GATA-binding protein 4, a member of the GATA transcription factor family, is associated with organ development in mesodermal and endodermal tissues, such as heart, gut, and gonads. The bipotential gonads emerge as genital ridges that originate from the proliferation of the coelomic epithelium, a process in which GATA4 is present. GATA4 is observed in ovarian somatic cells being expressed during the entire fetal period. In addition, GATA4 and GATA6 play a pivotal role in follicle assembly, granulosa cell differentiation, postnatal follicle growth, and luteinization. Conditional knockdown of these factors has led to female infertility at any stage of ovarian development [23]. Moreover, GATA factors have been associated with

ovarian development in different species including mammals, fish, birds, and fruit fly [23]. Some *in vitro* studies have shown that steroidogenic enzymes implicated in the synthesis of progesterone, androgens, and estrogens may be regulated by GATA4 and GATA6. In the absence of GATA4 and GATA6 expression, estradiol synthesis is impaired, and its inactivation is also stimulated by high expression of CYP1B1 [23]. Furthermore, GATA4 interacts with SF1/*NR5A1* and LRH1/*NR5A2* to activate the promoter of *3βHSD2*, *AMH*, *inhibin-α*, and *CYP19A1*, indicating GATA4 as a partner of these proteins in the regulation of ovarian steroidogeneses [23]. Interestingly, no pathogenic variants in *GATA4* have been reported in the POI phenotype, although heterozygous missense variants in this gene are associated with 46, XY DSD with or without congenital heart defects [51,52]. Herein, POI-45 carries a pathogenic homozygous missense variant in *GATA4* (c.1220C>G:p.Pro407Arg) (Table 2). She was diagnosed with primary amenorrhea, hearing loss, and additional kidney failure. Moreover, a congenital murmur was diagnosed at early age, and a surgery was done. Unfortunately, no clinical history of POI-45 congenital heart failure is available. In addition, a heterozygous missense variant in *GATA4* (c.280G>A:p.Ala94Thr) (Table 3) was also found in POI-12, who was diagnosed with secondary amenorrhea (Table 1). These new findings could expand our knowledge of *GATA4* in POI etiology.

## B) Meiosis and DNA repair genes (*MCM9, STAG3, SYCE1, CPEB1, UBR2, ATM, ERCC2, SYCP1, SMC1B*)

**MCM9.** Minichromosome maintenance complex component 9 is involved in homologous recombination (HR) during meiosis process. Null *MCM9* mice were sterile for showing defects in HR and gametogenesis [21]. Some homozygous and compound heterozygous variants were identified in different POI cohorts by using an NGS approach [21]. In our cohort, compound heterozygous missense variants in the *MCM9* [(c.2059T>C) and (c.3223C>T)] (Table 2) were found in POI-11, a patient presenting short stature and primary amenorrhea (Table 1). Indeed, the same phenotype was observed in the first report of this gene [18]. Unfortunately, chromosomal instability and segregation analyses in POI-11 could not be assessed. Moreover, twelve women with POI carried heterozygous variants in the *MCM9* gene, and most of these women were diagnosed as having secondary amenorrhea [20]. A rare and deleterious heterozygous variant in this gene (c.1163C>A;p.Thr388Asn) (Table 2) was also identified in POI-25 presenting with a mild phenotype characterized by secondary amenorrhea without short stature (Table 1) as previously described by Desai and collaborators [20]. It seems that *MCM9* may be causing POI in the autosomal dominant and recessive mode of inheritance.

**STAG3.** Two pathogenic heterozygous loss-of-function variants in *STAG3* were identified in one woman presenting with primary amenorrhea using whole-exome sequencing (Table 2) [26].

**SYCE1 and CPEB1.** These genes play a key role in meiosis by maintaining synaptonemal complex stability. In mouse studies, the impaired function of these genes has shown a reproductive phenotype of infertility due to the absence of synaptonemal complexes and meiotic arrest in meiosis I [53,54]. Pathogenic copy number variations (CNVs) and point variants in *SYCE1* and *CPEB1* genes are already implicated in the POI phenotype in distinct cohorts [12–15,17]. Jaillard and collarators reported a 123-kb duplication in *SYCE1* in one patient presenting with isolated POI [13]. Herein, POI-7 has a microduplication (11.4-kb) in this gene (Table 2). Furthermore, an 83.8-kb deletion in the *CPBE1* gene was found in POI-14 (Table 2), the same region previously identified in some POI cohorts [12,14,15]. In addition, the rare and deleterious heterozygous missense variant in *CPEB1* (c.259C>T;p.Arg87Cys) was also identified in POI-4 (Table 2). The 10q and 15q loci appear to be important POI loci, and these findings corroborate the role of *CPEB1* and *SYCE1* in POI etiology.

**UBR2.** *UBR2* encodes E3 ubiquitin ligase of the N-end rule proteolytic pathway for ubiquitin-mediated protein degradation. Null *Ubr2* mice showed chromosome fragility and impaired HR repair [55]. Moreover, these null mice were not viable, and $Ubr2^{+/-}$ showed reduced fertility [25]. POI-17, who was diagnosed with isolated POI presenting with secondary amenorrhea, carries a novel and pathogenic heterozygous missense variant in the *UBR2* gene (c.4843T>A;p.Ser1615Thr) (Tables 2 and 4). This is the first POI patient harboring a defect in the *UBR2* gene.

**ATM.** *ATM*, the first DNA repair gene associated with POI, is required for cell-cycle checkpoint [4]. Mutations in *ATM* led to syndromic POI, characterized by primary amenorrhea and associated with autosomal recessive ataxia telangiectasia [4]. POI-20 was diagnosed with primary amenorrhea, cubitus valgus, and tremor in her dominant hand without muscle atrophy (Table 1). She was identified with a compound heterozygous missense variant in the *ATM* gene, suggesting a recessive inheritance as previously described. Although severe cerebellar degeneration was not found in this patient, *ATM* variant features could lead to impaired DNA repair, reducing the germ cell pool of this patient. Interestingly, a rare homozygous missense (Table 3) was also identified in POI-24, a primary amenorrhea case with syndromic features, such as cubitus valgus and late psychomotor development (Table 1). However, POI-24 has either a homozygous frameshift insertion in the *FANCL* gene, which is also associated with chromosomal instability and Fanconi anemia. POI-24 has no Fanconi anemia features, however, although we were unable to rule out *FANCL* contribution in the POI phenotype of this patient.

**ERCC2.** The ERCC2 DNA repair factor is associated with complex phenotypes such as cerebrooculofacioskeletal syndrome 2 (MIM 610756), trichothiodystrophy 1 (MIM 601615), xeroderma pigmentosum D (MIM 278730). Female knock-in mice showed no signs of estrus cycle, small ovaries, and no preovulatory follicles. These mice also exhibited osteoporosis, kyphosis, osteosclerosis, early graying, cachexia, and reduced life span [43]. Herein, a rare heterozygous variant (c.1606G>A;p.Val536Met) in *ERCC2* (Table 3) was identified in POI-30, who was diagnosed with isolated primary amenorrhea (Table 1). In addition, this variant is predicted to be deleterious in all available *in silico* tools. Interestingly, three heterozygous pathogenic variants in the *ERCC6*, another member of the DNA repair cascade such as ERCC2, was found in one Chinese familial and sporadic case with POI [56]. Further studies may be needed to understand the contribution of *ERCC2* gene in POI; however, DNA repair genes have been strongly associated with POI etiology.

**SYCP1.** Although SYCP1 is an essential member of synaptonemal complex in labeling the axes of the chromosome during meiotic prophase I [22], no pathogenic variants associated with female infertility have been identified in these genes. $Sycp1^{-/-}$ male and female mice were infertile whereas $Sycp1^{+/-}$ mice were fertile [41]. Herein, POI-43, who presented secondary amenorrhea at 35 years of age (Table 1), has a heterozygous missense variant in *SYCP1* (c.1747C>G;p.Leu583Val), which is rare and deleterious in silico analysis (Table 3).

**SMC1B.** Structural Maintenance of Chromosomes 1B encodes a protein which belongs to the cohesin family, which is specific to the meiosis process [22]. Bouilly and collaborators [44] have reported two POI patients with different phenotypes harboring heterozygous variants in *SMC1B* in association with a second gene defect. One woman presenting primary amenorrhea had the p.Gln1177Leu heterozygous missense variant in *SMC1B* and the p.Ser5Arg heterozygous missense variant in the *BMP15* gene [44]. Moreover, a secondary amenorrhea case had one heterozygous missense variant in *SMC1B* (p.Ile221Thr) and one heterozygous variant in *NOBOX* (p.Gly91Trp, discussed above) [44]. In this current study, POI-41, who was diagnosed with secondary amenorrhea and isolated POI, carries two rare heterozygous missense variants in distinct genes, *SMC1B* and *HARS2* (Table 3). *HARS2* has been associated with Perrault

syndrome, which is an autosomal recessive disorder characterized by ovarian dysgenesis and deafness [22]. No deafness onset was found in POI-41 at 38 years of age. Interestingly, an affected sibling of POI-41, who was also diagnosed with secondary amenorrhea, carried both variants. DNA of the parents was unavailable for cosegregation analysis. Based on genetic features, these combined variants were classified as VUS (Table 3).

## C) X-chromosome genes: *NXF5, XPNPEP2, COL4A6*

The association of the X chromosome, region from Xq13.3 to Xq27, has been shown as a critical region for normal ovarian development [19]. Several genes disrupted by breakpoints in balanced X-autosome translocations have been associated with POI etiology, including *DIAPH2, XPNPEP2, DACH2, POF1B, CHM, PGRMC1, COL4A6,* and *NXF5* [19].

A cytogenetic analysis in a patient presenting with delay of puberty and primary amenorrhea, and no other clinical features showed a *de novo* translocation 46,XX, t(X;15)(Xq22;p11) with a breakpoint interval containing the *NXF5* gene [40]. Although the NXF5 function is not well known in ovary development, functional data demonstrated that the NXF5 protein is implicated in the posttranscriptional regulation of mRNA, and thus its heterozygous deficiency results in altered mRNA metabolism, similar to the proposed mechanism for the fragile-X-associated protein FMR1 [40]. Herein, three patients (POI-9, POI-10, POI-13) had *NXF5* variants combined with another variant in distinct genes (Table 3). POI-9 was diagnosed at 18 years of age with primary amenorrhea, puberty delay, and syndromic features, such as high arched palate, cubitus valgus, wide-spaced nipples, and short stature (Table 1). She carried two rare homozygous variants in the *NXF5* gene, one rare heterozygous variant in the *GATA4*, a rare heterozygous variant in the *NBN* gene, and one rare heterozygous variant in the *ATM* (Table 3). All variants were classified as VUS. The cytogenic analysis confirmed 46, XX karyotype and no deletions or duplications were found in this patient. A homozygous pathogenic variant in *NBN* was recently reported in an isolated POI case [4]; however, no heterozygous *NBN* and *ATM* variants were reported as POI cause, although they have been associated with cancer predisposition. It seems that POI is likely caused by the homozygous variants in the *NXF5*; however, we cannot eliminate an additional effect by other rare and deleterious heterozygous variants found in this patient. Moreover, a rare heterozygous nonsense variant in *NXF5* was found in POI-10, a secondary amenorrhea case. In addition, three missense variants in *COL4A6*, in *XPNPEP2*, and in *INHBB* were identified in this patient (Table 3). Inhibin, which belongs to the superfamily of TGF-β, acts as a negative regulator of FSH secretion, and impaired inhibin B bioactivity has shown increased susceptibility to POI [19]. POI-13 was diagnosed at 23 years of age presenting with primary amenorrhea and delay of puberty (Table 1). She carried two undescribed heterozygous missense variants, one in the *NXF5* and another one in the *SYCP1* (Table 3), a DNA repair gene discussed above. The mechanism of how these genes may contribute to POI in an oligogenic manner needed to be elucidated, and therefore, these variants were classified as VUS.

## Conclusion

The majority of the genetic etiology of POI remains uncharacterized in the literature. In this study, this gap has been narrowed with the massively parallel sequencing technique, which allowed us to expand genotype-phenotype correlations and to improve familial counseling, i.e., the identification of additional affected members at a younger age improves the quality of patients' lives, such as self-esteem in young women who had no breast development or planning for crypreservation of eggs for future intervention. Ultimately, a genetic diagnosis could better characterize the risk of developing underlying conditions, such as osteoporosis,

cardiovascular disease, allowing physicians to provide an efficient and appropriate hormone replacement therapy.

## Supporting information

**S1 Table. Analysis of identified variants *in silico* prediction tools.**
(DOCX)

## Acknowledgments

The authors thank the patients and their families for participating in this study. They are grateful to Ana Caroline Afonso, LIM42, and SELA teams for providing technical assistance.

## Author Contributions

**Conceptualization:** Monica M. França, Berenice B. Mendonca.

**Data curation:** Monica M. França.

**Formal analysis:** Monica M. França.

**Funding acquisition:** Berenice B. Mendonca.

**Investigation:** Monica M. França, Mariana F. A. Funari, Sorahia Domenice, Daniela R. Moraes, Everlayny F. Costalonga, Gustavo A. R. Maciel, Andrea T. Maciel-Guerra, Gil Guerra-Junior.

**Methodology:** Monica M. França, Mariana F. A. Funari, Mariza G. Santos, Mirian Y. Nishi.

**Software:** Antonio M. Lerario.

**Writing – original draft:** Monica M. França.

**Writing – review & editing:** Monica M. França, Mariana F. A. Funari, Berenice B. Mendonca.

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
