## [Decision Letter · Decision Letter 0]

12 May 2020

PONE-D-20-06248

Screening of Targeted Panel Genes in Brazilian Patients with Primary Ovarian Insufficiency

PLOS ONE

Dear Dr.Monica M França,

Thank you for submitting your manuscript to PLOS ONE. After careful consideration, we feel that it has merit but does not fully meet PLOS ONE’s publication criteria as it currently stands. Therefore, we invite you to submit a revised version of the manuscript that addresses the points raised during the review process.

The manuscript is interesting and relevant to the scientific community. 

The authors need to improve the description on how variants are charatarized as pathogenic and related to POI i.e. the utilization of predictors and other software. Please take into account the points addressed by the reviewers and correct accordingly.

We would appreciate receiving your revised manuscript by Jun 26 2020 11:59PM. To enhance the reproducibility of your results, we recommend that if applicable you deposit your laboratory protocols in protocols.io, where a protocol can be assigned its own identifier (DOI) such that it can be cited independently in the future. For instructions see: http://journals.plos.org/plosone/s/submission-guidelines#loc-laboratory-protocols

We look forward to receiving your revised manuscript.

Kind regards,

Klaus Brusgaard

Academic Editor

PLOS ONE

2. In your Methods section, please provide additional information about the participant recruitment method and the demographic details of your participants. Please ensure you have provided sufficient details to replicate the analyses such as: a) the recruitment date range (month and year),  b) a description of how participants were recruited, and c) descriptions of where participants were recruited and where the research took place.

3. Thank you for indicating that you had ethical approval for your study. In your Methods section, please ensure you have also stated whether you obtained consent from parents or guardians of the minors included in the study or whether the research ethics committee or IRB specifically waived the need for their consent.

"This work was supported by the Fundação de Amparo à Pesquisa do Estado de São Paulo (FAPESP) Grant 2014/14231-0 (to M.M.F.); FAPESP Grant 2013/02162-8, Nucleo de Estudos e Terapia Celular e Molecular (NETCEM), and Conselho Nacional de Desenvolvimento Científico e Tecnológico Grant 303002/2016-6 (to B.B.M.); and FAPESP Grant 2014/50137-5 (to SELA)."

"The authors received no specific funding for this work."

Reviewers' comments:

Reviewer's Responses to Questions

**Comments to the Author**

1. Is the manuscript technically sound, and do the data support the conclusions?

Reviewer #1: Yes

Reviewer #2: Yes

2. Has the statistical analysis been performed appropriately and rigorously? 

Reviewer #1: Yes

Reviewer #2: Yes

3. Have the authors made all data underlying the findings in their manuscript fully available?

Reviewer #1: Yes

Reviewer #2: No

4. Is the manuscript presented in an intelligible fashion and written in standard English?

Reviewer #1: Yes

Reviewer #2: Yes

5. Review Comments to the Author

Reviewer #1: Using a panel of candidate genes for targeted massively parallel sequencing (TMPS) and confirmation by Sanger sequencing, together with analyses of gene copy number variations (CNV), the authors investigated gene mutations in 50 POI patients. A genetic etiology was identified in 70% (36 of 50) of women with POI using the customized TMPS panel and a total of 24 pathogenic variants and 2 CNVs were identified in 48% (24 of 50) of POI patients. The study is well-designed and results add to the increasing literature regarding genetic basis of POI.

1. One of the difficulties of using exome sequencing to identify POI etiologies is the requirement of additional animal studies to confirm the significance of putative candidate genes with mutations. Several POI genes were identified here but have not previously reported in patients. The authors supported their conclusions based on familiar inheritance and/or mutant animal models. It is useful to point out the novel aspects of present findings in Discussion and summarize them in a Table by listing POI genes not previously reported in patients but have mutant animal findings or familiar inheritance to support their validity, such as BMP8B (BMP8B is essential for the generation of primordial germ cells in mice; Ying, 2000), SYCP1 (Mouse Sycp1 functions in synaptonemal complex assembly, meiotic recombination; de Vries, 2005), etc.

2. Genes in the X chromosome have been considered as hot spots for POI candidate genes. It is worthwhile to summarize or highlight these genes in the Table and discuss the implication of their chromosome location. Because of anticipated X chromosome inactivation and large number of X chromosome genes as POI candidates, is there any significance of different types of POI gene mutations in the sex chromosome?

Reviewer #2: Review of “Screening of Targeted Panel Genes in Brazilian Patients with Primary Ovarian Insufficiency” by Franca et al.

The authors analyzed the genomes of 50 patients with POI using targeted massively parallel sequencing to identify underlying mutations in candidate genes. Many cases of POI that are not associated with FMR1 mutations have an unknown genetic basis. This study chose candidate genes to sequence based on their involvement in ovarian development, folliculogenesis, meiosis and DNA repair. Abnormalities were identified in 70% of the patients. Many of the candidate genes have been implicated in POI or infertility in previous studies.

Major Points:

This is a thorough study of genetic mutations that may cause primary and secondary POI. Primary screening of target genes identified mutations in 16 genes. Mutations were then confirmed using Sanger sequencing.

I like that the authors break down the mutation analysis into groups of genes involved in different processes/pathways. However, the Discussion of all of the mutations is long and needs to be condensed with more of an overview of the findings. A diagram showing the number of mutations in each gene group might be helpful here. How many of the identified mutations occurred in genes that had never before been associated with POI? For instance, the SYCP1 mutation identified in this study appears to be the first mutation identified in this gene that is associated with POI. How many of the mutations were novel mutations in genes known to be associated with POI? For instance, the DAZL heterozygous variant (c.640C>T;p.Gln214*) is referred to as “undescribed”. I’m assuming this is therefore a novel mutation? These details could also be included in Table 2 with a column indicating “novel variant” or “previously identified”.

The characterization of pathogenic variants may need to be better described in the results. This appears to be based on prior identification of these variants and/or in silico predictions.

Please describe in more detail, the in silico characterization of the variants in the Methods and Results. Currently the Methods just has a list of pathogenicity prediction programs. Line 564 refers to “all available in silico tools” with no additional details.

If nearly all mutations occurred in previously identified genes, does that suggest that a paired down candidate gene approach could likely identify nearly all causative mutations in POI in the future? Do mutations in gene groups associate with phenotype (primary vs. secondary)?

In the Discussion, please expand upon this idea brought up in the abstract, “ A molecular etiology allowed us to establish better genotype-phenotype correlation and to improve familial counseling avoiding futures comorbidities.” Please say more about the genotype/phenotype correlation. How can the identification of genetic mutations benefit patients in the clinic?

Minor points:

The manuscript had spelling errors and grammar issues. I list here some, but not all, corrections. Please look through the manuscript and correct these issues.

line:

85 “usually diagnosed at a younger age”

89 “presenting as sporadic”

90-91 “is known to be autosomal dominant or autosomal recessive”

95 “several pathways such as gonadal development”

251 “POI is a heterogeneous disorder characterized by strong genetic basis that

252 comprises at least 75 genes

263 “and migrate to the genital ridge, giving rise to oocytes”

271 “POU5F1 was reported in one Chinese POI woman”

274 “19 year-old woman with secondary amenorrhea”

299 “infertile while heterozygous mice exhibit a subfertile phenotype.”

314 “ which is rare and deleterious in in silico analysis”

333 “Genes related to folliculogenesis”

350 “binding to the GDF9 promoter in vitro.”

357 “Further investigation”

393 “BMP15: Bone morphogenetic protein 15 has already been reported as an X-linked POI

416 “born from a consanguineous marriage,”

417 “a history of primary amenorrhea”

429 “ Drosophila, rodents, and human [27, 28].”

430 “Functional studies in Drosophila”

433 “was found in one woman”

6. PLOS authors have the option to publish the peer review history of their article (what does this mean?). If published, this will include your full peer review and any attached files.

Reviewer #1: No

Reviewer #2: Yes: Mara P. Steinkamp

---

## [Author Response · Author response to Decision Letter 0]

28 Jul 2020

June 2020 

Ref: Ms No.: PONE-D-20-06248

Prof. Dr. Klaus Brusgaard, 

Academic Editor 

PLOS ONE

Dear Prof. Brusgaard,

Thank you for reviewing our original manuscript entitled “Screening of Targeted Panel Genes in Brazilian Patients with Primary Ovarian Insufficiency” and for the review’s comments. 

We have changed the manuscript’s text in accordance to the comments made; all the changes we have made to the text are highlighted in yellow for review purposes. In addition, the manuscript was submitted to The American Journal Experts in order to be improved and all changes are highlighted in blue. 

The description on how variants were characterized and classified as pathogenic or variant of uncertain significance and related to POI were included in Table #2 and #3. In additional, a supplemental table #1 was added, which contains the analysis of in silico predictors’ software. In order to clarify it, we also addressed these issues in methods section. 

Nevertheless, we hope that we have addressed the issues raised and thank the reviewers for their constructive comments and interest in this manuscript. 

We look forward to your response in due course. 

Sincerely,

Monica Malheiros França, MSc, PhD 

Present address: The University of Chicago, Department of Medicine, Section of Endocrinology, 5841 S. Maryland Av., Chicago, IL, 60637, USA. 

Hospital das Clínicas, Laboratório de Hormônios e Genética Molecular

Av. Dr. Enéas de Carvalho Aguiar, 155, 2° andar, Bloco 6

CEP: 05403-900, São Paulo, Brasil.

Review comments to the Author:

Reviewer #1: Using a panel of candidate genes for targeted massively parallel sequencing (TMPS) and confirmation by Sanger sequencing, together with analyses of gene copy number variations (CNV), the authors investigated gene mutations in 50 POI patients. A genetic etiology was identified in 70% (36 of 50) of women with POI using the customized TMPS panel and a total of 24 pathogenic variants and 2 CNVs were identified in 48% (24 of 50) of POI patients. The study is well-designed and results add to the increasing literature regarding genetic basis of POI.

1. One of the difficulties of using exome sequencing to identify POI etiologies is the requirement of additional animal studies to confirm the significance of putative candidate genes with mutations. Several POI genes were identified here but have not previously reported in patients. The authors supported their conclusions based on familiar inheritance and/or mutant animal models. It is useful to point out the novel aspects of present findings in Discussion and summarize them in a Table by listing POI genes not previously reported in patients but have mutant animal findings or familiar inheritance to support their validity, such as BMP8B (BMP8B is essential for the generation of primordial germ cells in mice; Ying, 2000), SYCP1 (Mouse Sycp1 functions in synaptonemal complex assembly, meiotic recombination; de Vries, 2005), etc.

Response: We really appreciate your comment and suggestion. In order to incorporate your suggestion, we included the table #4, which highlights and correlates new genes/variants or inheritance related to POI patients identified in this manuscript. We also included a column of references in table #2 and #3 to make it clear for the readers. 

2. Genes in the X chromosome have been considered as hot spots for POI candidate genes. It is worthwhile to summarize or highlight these genes in the Table and discuss the implication of their chromosome location. Because of anticipated X chromosome inactivation and large number of X chromosome genes as POI candidates, is there any significance of different types of POI gene mutations in the sex chromosome?

Response: We thank the reviewer observation and suggestion. We included the chromosome position in the Supplemental table 4 and highlighted these genes in the new Figure 1, which was also suggested by reviewer#2. Of note, BMP15 is an X-chromosome gene that was included in the ovarian development subsection due to its well-established role in sheep, rodents, and POI patients. X-autosome translocation, and point mutations in X-chromosome genes have strongly been associated with ovarian dysgenesis and accelerated follicular atresia. Both syndromic and nonsyndromic patients have been described harboring these genetic defects related to X-chromosome, and it seems to be a hotspot between Xq13.3 to Xq27 as we briefly discussed (line 617-621). On top of that, these genes herein described, such as NXF5, COL4A6, and XPNPEP2, are related to POI, but their mechanisms have not been clearly elucidated in animal models as discussed in the manuscript. However, a combination of these three rare and deleterious variants in POI-10 may be causing her phenotype. Unfortunately, we were unable to evaluate a most likely X-chromosome inactivation in POI-10 due to a combination of her genetic defects. 

Reviewer #2: Review of “Screening of Targeted Panel Genes in Brazilian Patients with Primary Ovarian Insufficiency” by Franca et al.

The authors analyzed the genomes of 50 patients with POI using targeted massively parallel sequencing to identify underlying mutations in candidate genes. Many cases of POI that are not associated with FMR1 mutations have an unknown genetic basis. This study chose candidate genes to sequence based on their involvement in ovarian development, folliculogenesis, meiosis and DNA repair. Abnormalities were identified in 70% of the patients. Many of the candidate genes have been implicated in POI or infertility in previous studies.

MajorPoints:

This is a thorough study of genetic mutations that may cause primary and secondary POI. Primary screening of target genes identified mutations in 16 genes. Mutations were then confirmed using Sanger sequencing.

I like that the authors break down the mutation analysis into groups of genes involved in different processes/pathways. However, the Discussion of all of the mutations is long and needs to be condensed with more of an overview of the findings. A diagram showing the number of mutations in each gene group might be helpful here. How many of the identified mutations occurred in genes that had never before been associated with POI? For instance, the SYCP1 mutation identified in this study appears to be the first mutation identified in this gene that is associated with POI. How many of the mutations were novel mutations in genes known to be associated with POI? For instance, the DAZL heterozygous variant (c.640C>T;p.Gln214*) is referred to as “undescribed”. I’m assuming this is therefore a novel mutation? These details could also be included in Table 2 with a column indicating “novel variant” or “previously identified”.

Response: We really appreciate all comments and suggestions. A diagram showing the number of mutations in each group was included (named Figure 1). We understand that the Discussion might be long and we excluded some paragraphs, however we would like to request the reviewer`s permission to keep the same format, once we described a huge amount of data and POI shows a very heterogeneous genetic background. It would be very helpful for readers who are less familiar with the topic to connect literature information and results herein described. 

New columns were added in tables #2 and #3 (“Novel or previously reported variant in POI, ACMG classification and its criterion, and Supporting evidence related to POI in animal models and humans). Interestingly, all described variants in this study has not been reported, except for FSHR variant POI-38) and NOBOX variant (POI-35) previously reported by França et al (ref. 5) and Bouilly et al., (ref. 39), respectively. Moreover, an additional table (#4) was included in order to highlight the list of novel genes or mode of inheritance in POI patients supported by animal model findings that have not been described. 

The characterization of pathogenic variants may need to be better described in the results. This appears to be based on prior identification of these variants and/or in silico predictions.

Please describe in more detail, the in silico characterization of the variants in the Methods and Results. Currently the Methods just has a list of pathogenicity prediction programs. Line 564 refers to “all available in silico tools” with no additional details.

Response: We thank the reviewer`s request and all changes were made. A new column was added in tables #2 and #3 in regard to adding the ACMG classification and its criterion. Although some variants were classified as pathogenic and could be moved to the table #2, they have been kept in table #3, once this group of patients has harbored more than one variant in different genes. 

In order to clarify the in silico characterization, a new supplemental table #1 was included and details are available in Methods section (line 172-174). 

If nearly all mutations occurred in previously identified genes, does that suggest that a paired down candidate gene approach could likely identify nearly all causative mutations in POI in the future? Do mutations in gene groups associate with phenotype (primary vs. secondary)?

Response: We believe a customized panel of genes could be an efficiency approach for POI. First of all, this disease is highly heterogeneous and few mutations are associated with over 75 genes reported by several groups over the years and recently discussed by us (França and Mendonca, 2020 Ref#2). Secondly, TMPS approach would be very effective for barely screened cohort like ours, which wasn’t investigated by targeted genes using Sanger method. Finally, this approach could also allow us to screen huge genes that we have been unable to analyze it throughout the years. 

It is still unclear the association with primary vs. secondary amenorrhea and the group of genes described herein and in the literature. In fact, most of the genes implicated in POI are associated with both phenotypes in particular in the NGS era, which increased the number of solved cases. For instance, FSHR, GDF9, and BMP15, two well-known genes related to ovarian development, were found to cause primary and secondary amenorrhea. 

In the Discussion, please expand upon this idea brought up in the abstract, “ A molecular etiology allowed us to establish better genotype-phenotype correlation and to improve familial counseling avoiding futures comorbidities.” Please say more about the genotype/phenotype correlation. How can the identification of genetic mutations benefit patients in the clinic?

Response: We thank the reviewer suggestion and all changes were made as requested. Briefly, identifying the genetic cause of POI patients can have a benefit for additional affected members and their families. Moreover, knowing a genetic cause could enable early intervention with hormone replacement therapy to minimize comorbidities or cryopreservation of eggs to maximize future fertility potential. Ultimately, discovering new genes and or expanding pathways involved in the pathogenesis of POI enables insight into the important processes required for ovarian function and may highlight targets for new drug therapies and treatments.

Minor points:

The manuscript had spelling errors and grammar issues. I list here some, but not all, corrections. Please look through the manuscript and correct these issues.

line:

85 “usually diagnosed at a younger age”

89 “presenting as sporadic”

90-91 “is known to be autosomal dominant or autosomal recessive”

95 “several pathways such as gonadal development”

251 “POI is a heterogeneous disorder characterized by strong genetic basis that

252 comprises at least 75 genes

263 “and migrate to the genital ridge, giving rise to oocytes”

271 “POU5F1 was reported in one Chinese POI woman”

274 “19 year-old woman with secondary amenorrhea”

299 “infertile while heterozygous mice exhibit a subfertile phenotype.”

314 “ which is rare and deleterious in in silico analysis”

333 “Genes related to folliculogenesis”

350 “binding to the GDF9 promoter in vitro.”

357 “Further investigation”

393 “BMP15: Bone morphogenetic protein 15 has already been reported as an X-linked POI

416 “born from a consanguineous marriage,”

417 “a history of primary amenorrhea”

429 “ Drosophila, rodents, and human [27, 28].”

430 “Functional studies in Drosophila”

433 “was found in one woman”

Response: We thank the reviewer for all corrections. The changes were made and are highlighted in yellow. In addition, the manuscript was submitted to The American Journal Experts in order to be improved and all changes were highlighted in blue.

---

## [Decision Letter · Decision Letter 1]

5 Oct 2020

Screening of Targeted Panel Genes in Brazilian Patients with Primary Ovarian Insufficiency

PONE-D-20-06248R1

Dear Dr. Monica M França,

We’re pleased to inform you that your manuscript has been judged scientifically suitable for publication and will be formally accepted for publication once it meets all outstanding technical requirements.

Kind regards,

Klaus Brusgaard

Academic Editor

PLOS ONE

Additional Editor Comments (optional):

Reviewers' comments:

Reviewer's Responses to Questions

**Comments to the Author**

1. If the authors have adequately addressed your comments raised in a previous round of review and you feel that this manuscript is now acceptable for publication, you may indicate that here to bypass the “Comments to the Author” section, enter your conflict of interest statement in the “Confidential to Editor” section, and submit your "Accept" recommendation.

Reviewer #1: All comments have been addressed

Reviewer #2: All comments have been addressed

2. Is the manuscript technically sound, and do the data support the conclusions?

Reviewer #1: Yes

Reviewer #2: (No Response)

3. Has the statistical analysis been performed appropriately and rigorously? 

Reviewer #1: N/A

Reviewer #2: (No Response)

4. Have the authors made all data underlying the findings in their manuscript fully available?

Reviewer #1: Yes

Reviewer #2: (No Response)

5. Is the manuscript presented in an intelligible fashion and written in standard English?

Reviewer #1: Yes

Reviewer #2: (No Response)

6. Review Comments to the Author

Reviewer #1: (No Response)

Reviewer #2: (No Response)

7. PLOS authors have the option to publish the peer review history of their article (what does this mean?). If published, this will include your full peer review and any attached files.

Reviewer #1: No

Reviewer #2: **Yes: **Mara P. Steinkamp

---

## [Editor Report · Acceptance letter]

13 Oct 2020

PONE-D-20-06248R1 

Screening of Targeted Panel Genes in Brazilian Patients with Primary Ovarian Insufficiency 

Dear Dr. França:

I'm pleased to inform you that your manuscript has been deemed suitable for publication in PLOS ONE. Congratulations! Your manuscript is now with our production department. 

Kind regards, 

on behalf of

Dr. Klaus Brusgaard 

Academic Editor

PLOS ONE